# Hepatitis B screening and vaccination status of healthcare providers in Wakiso district, Uganda

Tonny Ssekamatte[1]*, Trasias Mukama[1], Simon P. S. Kibira[2], Rawlance Ndejjo[1], Justine Nnakate Bukenya[2], Zirimala Paul Alex Kimoga[3], Samuel Etajak[1], Rebecca Nuwematsiko[1], Esther Buregyeya[1], John C. Ssempebwa[1], John Bosco Isunju[1], Richard Kibirango Mugambe[1], Aisha Nalugya[1], Solomon Tsebeni Wafula[1], Joan Nankya Mutyoba[4]

1 Department of Disease Control and Environmental Health, Makerere University School of Public Health, Kampala, Uganda, 2 Department of Community Health and Behavioural Sciences, Makerere University School of Public Health, Kampala, Uganda, 3 Department of Health, Wakiso District Local Government, Wakiso, Uganda, 4 Department of Biostatistics and Epidemiology, Makerere University School of Public Health, Kampala, Uganda

* ssekamattet.toca@gmail.com, tssekamatte@musph.ac.ug

**Data Availability Statement:** All relevant data are within the paper and its Supporting Information files.

## Abstract

### Background

Screening and vaccination against Hepatitis B virus (HBV) infection remains the most effective intervention in curbing the disease. However, there is limited evidence on the factors associated with the uptake of these services in Uganda. This study determined the uptake of HBV screening and vaccination status, and associated factors among Healthcare Providers (HCPs) in Wakiso district, Uganda.

### Materials and methods

This cross-sectional study was conducted among 306 HCPs, randomly selected from 55 healthcare facilities. Prevalence ratios (PR) were used to determine the factors associated with HBV screening and vaccination status of HCPs.

### Results

Of the 306 HCPs, 230 (75.2%) had ever screened for HBV infection while 177 (57.8%) were fully vaccinated. Being male was positively associated with 'ever been screened' for HBV infection (Adjusted PR = 1.27, 95%CI 1.13–1.41). Working in a public healthcare facility (Adjusted PR = 0.78, 95%CI 0.68–0.90) was negatively associated with ever been screened. Male sex (Adjusted PR = 1.21, 95%CI 1.01–1.46), the belief that the HBV vaccine was safe (Adjusted PR = 1.72, 95%CI 1.03–2.89) and ever been screened (Adjusted PR = 2.28, 95%CI 1.56–3.34) were positively associated with being fully vaccinated. However, working in a public healthcare facility (Adjusted PR = 0.79, 95%CI 0.64–0.98), self-perceived risk of HBV infection (Adjusted PR = 0.72, 95% CI:0.62–0.84), and working in a

**Funding:** The author(s) received no specific funding for this work.

**Competing interests:** The authors have declared that no competing interests exist.

healthcare facility with infection control guidelines (Adjusted PR = 0.79, 95%CI 0.66–0.95) were negatively associated with being fully vaccinated.

## Conclusion

Three quarters of HCPs had ever been screened for HBV while slightly more than half were fully vaccinated. HBV screening and vaccination interventions need to consider the HCP sex, risk perception, attitude towards safety and efficacy of the hepatitis B vaccine, and healthcare facility characteristics such as ownership and availability of infection control guidelines, in order to be successful.

## Background

Hepatitis B virus (HBV) infection remains a significant global health challenge, causing substantial morbidity and mortality [1, 2]. In 2015, the World Health Organization (WHO) reported over 257 million cases and 887,000 HBV infection- related deaths [1]. The burden of HBV infection is higher in the WHO African Region and Western Pacific Region with an adult prevalence of 6.1% and 6.2% respectively [1]. In Uganda, the adult prevalence of HBV infection stands at 4.1% (5.4% among men and 3.0% among women) [3]. Although there is limited evidence on the burden of HBV among healthcare providers (HCPs), Ziraba, Bwogi [4] reported a seroprevalence of 8.1% and a lifetime exposure prevalence of 48.1% among HCPs in a tertiary hospital in Uganda.

HCPs are at an elevated risk of contracting HBV infection due to their frequent occupational exposure to infected blood and other body fluids [1, 5, 6]. Indeed, HCPs have an up to four-fold increased risk of acquiring HBV infection compared to the general population [7, 8]. HBV infections in healthcare settings usually result from needle prick injuries and failure to adhere to universal infection prevention precautions [9]. The public health challenge of HBV infections in healthcare settings in many resource-constrained countries is accelerated by limited access to diagnosis and treatment for hepatitis B [1, 10]. In addition, a lack of awareness of HBV and precautionary measures to take against blood-borne infections among HCPs has been reported to be a risk factor for infection [11–13]. Yet, if left untreated, chronic HBV carriers develop severe complications including cirrhosis, liver failure and hepatocellular carcinoma [1]. These outcomes pose a serious burden not only on the healthcare system but also at household level [14–16].

The WHO recommends that high-risk groups such as HCPs should be screened for and vaccinated with three doses of the recombinant deoxyribonucleic acid (DNA) or plasma-derived hepatitis B vaccine [17]. HBV screening and diagnosis are vital for access to appropriate prevention, care and treatment services [2]. Early identification of persons with chronic HBV infection enables possible receipt of necessary care and treatment to prevent or delay the progression of liver disease [2]. Screening also provides an opportunity to link people to interventions to reduce transmission through counselling on risk behaviours and hepatitis B vaccination [18].

Given the public health threat posed by HBV infection, in 2014, the Ugandan Ministry of Health developed statutory instruments that declared HBV infection a public health threat and made the vaccination of HCPs mandatory [19, 20]. Despite the existence of these statutory instruments, and safe and effective screening and vaccination services, screening and uptake of these HBV vaccinations among HCPs in the developing world remain low [21]. As a result,

many HCPs normally present with advanced stages of the disease because they are not aware of their status [22]. The low uptake of HBV vaccination services among HCPs in sub-Saharan Africa has been attributed to the cost of the vaccine and lack of awareness of its availability [23]. This is coupled with limited evidence of screening and uptake of hepatitis B vaccination among HCPs [24], and a knowledge gap on the factors associated with HBV screening and completion of the hepatitis B vaccination schedule. This study was premised on the Knowledge, Attitude and Practice (KAP) model to determine factors associated with HBV screening and vaccination status of HCPs in Wakiso district, Uganda. This model has previously been used in similar studies to understand the screening and vaccination behaviour of HCPs [25–27].

## Materials and methods

### Study setting and population

Data were collected from HCPs in the seven health sub-districts of Wakiso district; Busiro East, Busiro North, Busiro South, Kyadondo East, Kyadondo North, Kyadondo South and Entebbe Municipality. Health sub-districts are the next immediate level of health administration in a district monitoring delivery of health services at the sub-county levels, usually at health centre IIIs and IIs. Wakiso district is located in the central region of Uganda and partly encircles Kampala capital city. The district has a total of 589 health care facilities (including privately owned healthcare facilities), of which 15 are general hospitals, 19 health centre IVs, 165 health centre IIIs and 153 health centre IIs, and the remainder are clinics [28]. Uganda's healthcare system is decentralised with higher levels, providing additional services to what is offered at their preceding level. The health centre IIs have a target population of 5,000 people, and are mandated to provide preventive, promotive and outpatient services, outreach care and emergency services; health centre IIIs have a target of 20,000 people and in addition provide maternity, inpatient and laboratory services. Health centre IVs have a target population of 100,000 and provide emergency surgery, blood transfusion and laboratory as additional services. General hospitals have a target population of 500,000 and their additional services are in-service training, consultation and research [28].

### Study design, sample size and sampling

A cross-sectional study design was used with data collected in July 2018. The sample size for this study was determined using the Kish Leslie formula for cross-sectional studies. Considering a prevalence (p) of completion of three hepatitis B vaccine doses of 25.6% [29] at a 95% level of confidence and a margin of error of 0.05, a minimum sample size of 293 HCPs was obtained. The estimated sample size was adjusted using a design effect of 1.2 and a non-response rate of 10% [20, 30], yielding a final sample size of 325 HCPs. A total of 6 general hospitals and 16 health centre IVs were purposively selected while 33 health centre IIIs were randomly selected from the Wakiso district healthcare facility inventory. General hospitals and health centre IVs included private for profit, private not for profit and public healthcare facilities. The rationale for the selection of health centre IVs and general hospitals was because they serve the greater majority of the population and offer high-risk medical interventions such as caesarean deliveries and blood transfusion. Such interventions escalate the risk of HBV infection among HCPs. The number of HCPs selected at each healthcare facility was dependent on the number of staff employed at the healthcare facility and their availability during the time of the survey. Random sampling was applied in the selection of HCPs in each healthcare facility.

## Study variables

The outcome variables were vaccination status (categorized as either being fully vaccinated against HBV infection or not fully vaccinated) and "ever been screened" for hepatitis B. A HCP was considered fully vaccinated if they had taken all three recommended vaccine doses [6] with those who reported zero to two vaccine doses considered not to be fully vaccinated [17]. The independent variables included HCP knowledge, attitude and practices and socio-demographic characteristics such as age, main department of work, cadre, highest level of education, location of health facility (rural vs urban), history of injury, and years of work experience as HCP.

## Data collection tool, data management and statistical analysis

A structured questionnaire was used to collect data on screening and HCP vaccination status. Data were entered using Android enabled mobile phones and tablets. These were loaded with the KoboCollect application, and data were synchronised onto the server daily. Mobile data collection allowed for real-time data capture and entry, minimised errors at entry and eased data cleaning. To ensure that the data were secure, only the principal investigators had the security key to the KoboCollect server, where the data were being sent during data collection. The data collection tool was evaluated for face and internal validity by a team of experts in hepatitis B research who are based at the Makerere University College of Health Sciences. It was later piloted among 10 healthcare providers in a Kampala public healthcare facility. To enhance data quality, research assistants were trained and the data collection tool was pretested.

Data were analysed using Stata 14.0 statistical software (StataCorp Texas, USA). Descriptive analyses such as frequencies, proportions, and means (*where appropriate*) were performed for HCPs demographic characteristics, and their knowledge and attitudes regarding hepatitis B. The outcome variables (ever screened for hepatitis B and vaccination status) were dichotomous with options Yes coded 1 and No coded 0. To assess the association between the outcome variables and each explanatory variable, we considered a generalized linear model of the Poisson family, with logarithm as the canonical link function and applying robust error variance which provided crude prevalence ratios (PRs) and their corresponding 95% confidence intervals (CIs). Variables with a threshold probability (*p*) value $\leq 0.2$ in bivariable models were all added into the multivariable regression model and a stepwise backward elimination method used until only significant predictors were retained in the model. Both crude PRs and adjusted PRs have been reported. All *p*-values were two-sided and considered significant if less than 0.05.

## Ethics statement

Ethical approval was obtained from Makerere University School of Public Health Higher Degrees Research and Ethics Committee. Administrative clearance was sought from Wakiso district Local government and the management of the participating healthcare facilities. Written informed consent was obtained from all the HCPs who participated in the study.

## Results

### Demographic characteristics of the participants

Of the 306 HCPs that participated in the study (response rate was 94.1%), 206 (67.3%) were female, 186 (60.8%) were aged below 30 years with an average age of 29.5 years (SD ± 7.7) and 204 (66.7%) worked at health facilities in urban areas. HCPs had an average of 5.7 (SD ± 5.6) years of experience (Table 1).

**Table 1. Socio-demographic characteristics of the study participants.**

| Variable | Category | n (N = 306) | Percentage (%) |
|---|---|---|---|
| Location of health facility | Rural | 102 | 33.3 |
| | Urban | 204 | 66.7 |
| Sex | Female | 206 | 67.3 |
| | Male | 100 | 32.7 |
| Age in years | < 30 | 186 | 60.8 |
| | ≥ 30 | 120 | 39.2 |
| Marital status | Married | 128 | 41.8 |
| | Not married | 178 | 58.2 |
| HCP cadre | | | |
| | Clinical officer /general practitioners | 82 | 26.8 |
| | Nurses / midwives | 109 | 35.6 |
| | Lab personnel | 85 | 27.8 |
| | Anaesthetists | 30 | 9.8 |
| Years of experience as HCP | < 5 | 173 | 56.5 |
| | ≥ 5 | 133 | 43.5 |
| Level of health care facility | Health centre III | 133 | 43.5 |
| | Health centre IV | 120 | 39.2 |
| | Hospital | 53 | 17.2 |
| Ownership of facility | Private | 136 | 44.4 |
| | Private not for profit | 30 | 9.8 |
| | Public | 140 | 45.7 |

## Hepatitis B screening and associated factors among HCPs

Out of 306 HCPs, 230 (75.2%) had ever been screened for Hepatitis B. There were significant differences in screening for HBV across sociodemographic characteristics. About 68% of the female HCPs had ever screened for HBV compared to 90% of the male HCPs (p-value < 0.001). Approximately 85% of the HCPs in private health facilities had ever been screened for HBV compared to only 62.9% in public health facilities (p-value < 0.001).

At bivariate analysis, participants were more likely to have ever screened for HBV if they were males, worked in hospitals as compared to lower-level health centre IIIs or worked in a health centre located in an urban setting. However, participants in public facilities were less likely to screen for HBV compared to those in privately owned facilities. As regard to knowledge, a higher proportion (83.9%) of those who had heard about post-exposure prophylaxis (PEP) for hepatitis B had screened compared to 71.4% of HCPs who had not heard about PEP for HBV infection. In multivariable regression, male HCPs had a 27% higher likelihood of ever been screened for hepatitis B compared to female HCPs (Adjusted PR = 1.27, 95% CI: 1.13–1.41). HCPs in public healthcare facilities had a 22% lower likelihood of ever been screened for hepatitis B compared to those from private healthcare facilities (Adjusted PR = 0.78, 95% CI:0.68–0.90). (Table 2).

## Hepatitis B vaccination status of HCPs

A total of 177 (57.8%) HCPs were fully vaccinated against HBV (had received three doses of HBV vaccine). About 41 (13.4%) had received only two doses, 24 (7.8%) only one dose, while 64 (20.9%) had not received any vaccination for HBV at all. About 48.7% (39/76) of the HCPs had received at least one HBV vaccine dose without ever being screened. About 11.7% (27/230) of HCPs who had ever been screened for HBV had not received any HBV vaccine dose.

**Table 2. Factors associated with "ever been screened" for HBV infection among healthcare providers in Wakiso district, central Uganda.**

| Variable | Ever been screened for HBV | | Crude PR (95% CI) | p-value | Adjusted PR (95% CI) | p-value |
|---|---|---|---|---|---|---|
| | No | Yes | | | | |
| **Sex** | | | | | | |
| Female | 66 (32.0) | 140 (68.0) | 1 | | 1 | |
| Male | 10 (10.0) | 90 (90.0) | 1.32 (1.18–1.48) | < **0.001** | 1.27 (1.13–1.41) | < **0.001** |
| **Cadre** | | | | | | |
| Clinical officer /GP | 21 (25.6) | 61 (74.4) | 1 | | | |
| Nurses / midwives | 27 (24.8) | 82 (75.2) | 1.01 (0.86–1.19) | 0.895 | | |
| Lab personnel | 18 (21.2) | 82 (78.8) | 1.06 (0.89–1.25) | 0.501 | | |
| Anaesthetists | 10 (33.3) | 20 (66.7) | 0.90 (0.68–1.19) | 0.449 | | |
| **Years of experience** | | | | | | |
| < 5 years | 39 (22.5) | 134 (77.5) | 1 | | | |
| ≥ 5 years | 37 (27.8) | 96 (72.2) | 0.93 (0.82–1.06) | 0.298 | | |
| **Healthcare level** | | | | | | |
| Health centre II-III | 38 (28.6) | 95 (71.4) | 1 | | | |
| Health centre IV | 31 (25.8) | 89 (74.2) | 1.04 (0.89–1.21) | 0.625 | | |
| Hospital | 07 (13.2) | 46 (86.8) | 1.22 (1.05–1.41) | **0.011** | | |
| **Ownership of facility** | | | | | | |
| Private | 21 (15.4) | 115 (84.6) | 1 | | 1 | |
| PNFP | 03 (10.0) | 27 (9.0) | 1.06 (0.93–1.22) | 0.381 | 0.99 (0.86–1.15) | 0.909 |
| Public | 52 (37.1) | 88 (62.9) | 0.74 (0.64–0.86) | < **0.001** | 0.78 (0.68–0.90) | **0.001** |
| **Location** | | | | | | |
| Rural | 36 (35.3) | 66(64.7) | 1 | | 1 | |
| Urban | 40 (19.6) | 164 (80.4) | 1.24 (1.06–1.46) | **0.007** | 1.15 (0.98–1.34) | 0.089 |
| **Age in years** | | | | | | |
| < 30 | 40 (21.5) | 146 (78.5) | 1 | | | |
| ≥ 30 | 36 (30.0) | 84 (70.0) | 0.89 (0.78–1.03) | 0.107 | | |
| **Marital status** | | | | | | |
| Married | 38 (29.7) | 90 (70.3) | 1 | | | |
| Not married | 38 (21.3) | 140 (78.7) | 1.12 (0.97–1.28) | 0.107 | | |
| **Knowledge and attitudes** | | | | | | |
| **HBV infection can be transmitted by carriers** | | | | | | |
| No | 15 (30.6) | 34 (69.4) | 1 | | | |
| Yes | 61 (23.7) | 196 (76.3) | 1.10 (0.90–1.34) | 0.351 | | |
| **Perceived HBV risk** | | | | | | |
| No | 1 (7.1) | 13 (92.8) | 1 | | 1 | |
| Yes | 75 (25.7) | 217 (74.3) | 0.80 (0.68–0.94) | **0.007** | 0.84 [0.71–1.00] | 0.052 |
| **Thought they were at high risk** | | | | | | |
| No | 13 (27.7) | 34 (72.3) | 1 | | | |
| Yes | 62 (25.3) | 183 (74.7) | 1.03 (0.85–1.25) | 0.743 | | |
| **Job increases risk** | | | | | | |
| No | 1 (20.0) | 4 (80.0) | 1 | | | |
| Yes | 75 (24.9) | 226 (75.1) | 0.94 (0.60–1.46) | 0.779 | | |
| **Heard about Hepatitis B Immuno-response test** | | | | | | |
| No | 61 (26.4) | 170 (73.6) | 1 | | | |
| Yes | 15 (20.0) | 60 (80.0) | 1.09 (0.95–1.25) | 0.233 | | |
| **Heard about PEP** | | | | | | |

*(Continued)*

**Table 2.** (Continued)

| Variable | Ever been screened for HBV | | Crude PR (95% CI) | p-value | Adjusted PR (95% CI) | p-value |
|---|---|---|---|---|---|---|
| | No | Yes | | | | |
| No | 61 (28.6) | 152 (71.4) | 1 | | | |
| Yes | 15 (16.1) | 78 (83.9) | 1.18 (1.04–1.33) | **0.010** | | |
| **Infected HCPs may infect patients** | | | | | | |
| No | 4 (36.4) | 7 (63.6) | 1 | | | |
| Yes | 72 (24.4) | 223 (75.6) | 1.19 (0.76–1.87) | 0.455 | | |
| **IC\* guidelines** | | | | | | |
| No | 07 (18.0) | 32 (82.0) | 1 | | | |
| Yes | 69 (25.8) | 198 (74.2) | 0.90 (0.77–1.06) | 0.225 | | |

GP = general practitioner. PEP = Post-exposure prophylaxis. IC* = had infection control guidelines at the health facility

At bivariate analysis, the proportion of male HCPs who had completed HBV vaccination was 32% higher than that of female counterparts. HCPs working in an urban setting had a 34% higher likelihood of being fully vaccinated against HBV infection as compared to those working in rural health facilities. The proportion of HCPs in public health facilities that had been fully vaccinated was 35% lower than HCPs in private health facilities. Additionally, HCPs who believed that the HBV vaccine was safe or effective in preventing the disease were more likely to be fully vaccinated against HBV infection. Considering oneself to be at risk of HBV and working in a health facility with infection control guidelines were negatively associated with being fully vaccinated against HBV infection.

After adjusting for potential confounders, screening for HBV (Adjusted PR = 2.28, 95% CI: 1.56–3.34) and believing that the HBV vaccine is safe (Adjusted PR = 1.72, 95% CI: 1.03–2.89) were positively associated with completion of the HBV vaccination schedule. Conversely, working in a public vs private health facility (Adjusted PR = 0.79, 95% CI: 0.64–0.98), working in a facility with infection control guidelines (Adjusted PR = 0.79, 95% CI: 0.66–0.95) and considering oneself to be at risk of HBV (Adjusted PR = 0.72, 95% CI: 0.62–0.84) were negatively associated with completion of the HBV vaccination schedule (Table 3).

## Discussion

Our study determined Hepatitis B screening and vaccination status of HCPs and associated factors in Wakiso district, Uganda. The study found that one-quarter of HCPs had never screened for HBV while nearly four in 10 were not fully vaccinated. We also found that male HCPs and those working in private healthcare facilities were more likely to report having been screened for HBV. Relatedly, believing in the efficacy and safety of the vaccine and working in a private healthcare facility were associated with being fully vaccinated. However, HCPs who worked in a facility with infection control guidelines, those who perceived themselves to be at risk of hepatitis B infection and perceived their job as a risk for infection were less likely to report full vaccination status.

A quarter of HCPs in our study had never been screened for HBV infection, thus not meeting the WHO and Uganda MOH recommendations that all health workers should be screened. Nevertheless, the proportion of HCPs who knew their HBV status in our study was higher than that reported among HCPs in two northern Tanzanian hospitals [31]. Potential reasons for non-screening include the inadequacy of required supplies such as kits and the low levels of knowledge on HBV among HCPs [32]. Regarding vaccination, we found that 57.8% of

**Table 3. Factors associated with vaccination status of healthcare providers in Wakiso district, central Uganda.**

| Variable | Vaccination status n (%) | | Crude PR (95% CI) | p-value | Adjusted PR (95% CI) | p-value |
|---|---|---|---|---|---|---|
| | Not fully vaccinated | Fully vaccinated | | | | |
| **Sex** | | | | | | |
| Female | 98 (47.6) | 108 (52.4) | 1 | | 1 | |
| Male | 31 (31.0) | 69 (69.0) | 1.32 (1.09–1.58) | **0.004** | 1.21 (1.01–1.46) | **0.035** |
| **Cadre** | | | | | | |
| Clinical officer /GP | 46 (42.2) | 63 (57.8) | 1 | | | |
| Nurses / midwives | 33 (40.2) | 49 (59.8) | 0.97 (0.76–1.23) | 0.785 | | |
| Lab personnel | 41 (48.2) | 44 (51.8) | 0.86 (0.66–1.14) | 0.301 | | |
| Anaesthetists | 9 (30.0) | 21 (70.0) | 1.17 (0.87–1.57) | 0.292 | | |
| **Years of experience as HCP** | | | | | | |
| < 5 years | 68 (39.3) | 105 (60.7) | 1 | | | |
| ≥ 5 years | 61 (45.9) | 72 (54.1) | 0.89 (0.73–1.09) | 0.256 | | |
| **Healthcare level** | | | | | | |
| Health centre II-III | 64 (48.1) | 69 (51.9) | 1 | | | |
| Health centre IV | 47 (39.2) | 73 (60.8) | 1.17 (0.94–1.46) | 0.152 | | |
| Hospital | 18 (34.0) | 35 (66.0) | 1.27 (0.99–1.64) | 0.062 | | |
| **Ownership of facility** | | | | | | |
| Private | 44 (32.3) | 92(67.7) | 1 | | 1 | |
| PNFP | 07 (23.3) | 23 576.7) | 1.13 (0.90–1.43) | 0.285 | 1.24 (0.96–1.60) | 0.107 |
| Public | 78 (55.7) | 62 (44.3) | 0.65 (0.53–0.82) | **< 0.001** | 0.79 (0.64–0.98) | **0.033** |
| **Location** | | | | | | |
| Rural | 54 (52.9) | 48 (47.1) | 1 | | | |
| Urban | 75 (36.8) | 129 (63.2) | 1.34 (1.07–1.69) | **0.012** | | |
| **Age in years** | | | | | | |
| < 30 | 73 (39.3) | 113 (60.7) | 1 | | | |
| ≥ 30 | 56 (46.7) | 64 (53.3) | 0.88 (0.72–1.08) | 0.210 | | |
| **Marital status** | | | | | | |
| Married | 62 (48.4) | 66 (51.6) | 1 | | | |
| Not married | 67 (37.6) | 111 (62.4) | 1.21 (0.99–1.48) | 0.067 | | |
| **Ever been screened for HBV** | | | | | | |
| No | 56 (73.7) | 20 (26.3) | 1 | | 1 | |
| Yes | 73 (31.7) | 157 (68.3) | 2.59 (1.76–3.82) | **< 0.001** | 2.28 (1.56–3.34) | **< 0.001** |
| **Knowledge and attitude** | | | | | | |
| **HBV can be transmitted by carriers** | | | | | | |
| No | 20 (40.8) | 29 (59.2) | 1 | | | |
| Yes | 109(42.4) | 148 (57.6) | 0.97 (0.75–1.26) | 0.834 | | |
| **Perceived HBV risk** | | | | | | |
| No | 1 (7.1) | 13 (92.9) | 1 | | 1 | |
| Yes | 128 (43.8) | 164 (56.2) | 0.60 (0.51–0.72) | **< 0.001** | 0.72 (0.62–0.84) | **< 0.001** |
| **Job increases risk** | | | | | | |
| No | 1 (20.0) | 4 (80) | 1 | | | |
| Yes | 128 (42.5) | 173 (57.5) | 0.72 (0.46–1.13) | 0.149 | | |
| **Infected HCPs may infect patients** | | | | | | |
| No | 4 (36.4) | 7 (63.6) | 1 | | | |
| Yes | 125 (42.4) | 170 (57.6) | 0.91 (0.57–1.43) | 0.671 | | |
| **Hepatitis B vaccine is very effective** | | | | | | |
| No | 47 (61.0) | 30 (39.0) | 1 | | | |

*(Continued)*

**Table 3.** (Continued)

| Variable | Vaccination status n (%) | | Crude PR (95% CI) | p-value | Adjusted PR (95% CI) | p-value |
|---|---|---|---|---|---|---|
| | Not fully vaccinated | Fully vaccinated | | | | |
| Yes | 82 (35.8) | 147 (64.2) | 1.65 (1.23–2.22) | **0.001** | | |
| **Hepatitis B vaccine is safe** | | | | | | |
| No | 16 (66.7) | 8 (33.3) | 1 | | 1 | |
| Yes | 113 (40.1) | 169 (59.9) | 1.80 (1.01–3.19) | **0.045** | 1.72 (1.03–2.89) | **0.039** |
| **IC\* guidelines** | | | | | | |
| No | 08 (20.5) | 31 (79.5) | 1 | | 1 | |
| Yes | 121 (45.3) | 146 (54.7) | 0.69 (0.56–0.83) | **< 0.001** | 0.79 (0.66–0.95) | **0.011** |

GP = general practitioner. PEP = Post-exposure prophylaxis. IC\* = had infection control guidelines at the health facility

HCPs had been fully vaccinated, which proportion among HCPs is still low and falls short of the global target of reducing HBV infections to 0.1% by 2030 [33]. Vaccination completion rates below 100% expose HCPs and their patients to a risk of contracting the HBV infection and spreading it further in the population. Previous studies in Uganda have reported single-dose vaccination proportions ranging from 5.1% in 2006 [34] to 78% among major hospitals in the Capital in 2014 [24] and thus there has been an improvement in the HCP vaccination status. Contributory factors to this improvement include hepatitis B vaccination campaigns and the development of statutory instruments requiring mandatory vaccination of all HCPs within 6 weeks of commencement of clinical work by the Ministry of Health, and increased public awareness [19]. The completion rate in this study compares to studies from Tanzania, Nigeria, China and Rwanda, where 48% to 90% of HCPs had completed the hepatitis B vaccination schedule [7–9, 35]. The first step should be to bridge the existing gap between screening and vaccination and ensure that all HCPs who screen for HBV and are eligible for the vaccine obtain it which will require an increased access to screening and vaccination services.

In our study, male HCPs were more likely to have ever screened and be fully vaccinated compared to their female counterparts. This could be so because, in the Ugandan health system, males are usually of higher cadres and take on more administrative roles and thus more likely to encounter opportunities for screening and vaccination. This finding contrasts with other studies where males were less likely to be vaccinated [20]. Moreover, from the general population, females have usually been found to have better health seeking behaviours compared to males (UBOS, 2012).

Our study revealed a marginal association between the location of the healthcare facilities where the HCP was working and screening for HBV. HCPs in healthcare facilities in an urban setting were more likely to have screened compared to their counterparts in rural settings. Yuan, Wang [35] revealed similar findings. This could be because of the availability of services in urban settings compared to rural settings [36]. Moreover, hepatitis B screening services are mainly in urban settings, which poses a cost implication to those working in rural areas who have to incur transport and vaccination costs to access the services from private healthcare facilities. Our findings henceforth, emphasize the need to extend screening services to HCPs, especially those in rural areas where access to services is lower.

HCPs who were working in public health facilities at the time of the survey were less likely to have been screened for HBV infection compared to those in private health facilities. This may be because Hepatitis B services are more available in private healthcare facilities compared to public facilities. Private healthcare facilities have taken on provision of screening and vaccination services as a business hence have to ensure continuous availability of supplies whereas

in public facilities, availability is heavily dependent on stocking by the government medicine supply agency and stockouts are frequently experienced. More so, some private hospitals offer their HCPs screening and vaccination services at a subsidised cost, which is currently not the case in most public healthcare facilities in Uganda. Decentralisation of HBV screening and vaccination services to lower-level public healthcare facilities is likely to solve this challenge.

HCPs working in a health facility with infection control guidelines were less likely to report being fully vaccinated against hepatitis B. Existence of infection control protocols could influence the belief that these are sufficient to safeguard HCPs against HBV infection, especially if the same HCPs have a negative attitude toward the vaccines. The low proportion of HCPs who were fully vaccinated in healthcare facilities with infection control guidelines could also have been due to the negative attitudes about the safety and effectiveness of the vaccine, for example, a 2018 widespread sale of suspected falsified hepatitis B vaccines [37]. In addition, without access to vaccination services, a positive attitude may not be enough for the HCPs. Therefore, vaccination programs among HCPs in Uganda ought to rebuild HCP's trust in hepatitis B vaccines and make vaccination services accessible as well. In addition, these need to reemphasize that following infection control protocols and other standard operating procedures in healthcare facilities is not enough for HCPs to shun vaccination as their risk to HBV infection remains considerably high.

## Strengths and limitations of the study

This is one of the few studies that has established the level of hepatitis B screening, completion of the hepatitis B vaccination schedule, and associated factors among HCPs in public and private health facilities in Uganda. Since HBV vaccination is a key event in a HCP's professional life, it is less likely that they would forget if they had ever screened for HBV infection or completed the hepatitis B vaccination schedule thus minimising recall bias. However, our findings may suffer from social desirability bias as HCPs may have been more likely to report favourable health access and utilisation. This being a cross-sectional study, it can only provide associations, other than causality about factors leading to screening for hepatitis B and vaccination status. These findings may not also be generalisable for all HCPs in Uganda since interventions among HCPs may have been implemented in higher risk regions such as the northern and eastern parts of the country. Wakiso district is also more cosmopolitan in Uganda and may have unique exposures to information compared to upcountry districts.

## Conclusion and recommendations

Both HBV screening and vaccination completion rates were low among HCPs in Wakiso, a cosmopolitan district in central Uganda. Being male was positively associated with ever being screened for HBV infection while working in a public healthcare facility was negatively associated with screening for HBV infection. This indicates the need for targeted HBV screening for HCPs working in public healthcare facilities. Having infection control guidelines at the health facility and the HCP belief of being at risk of HBV infection was not associated with completion of the hepatitis B vaccination schedule. However, the belief that the vaccine was safe was associated with completion of the HBV vaccination schedule. Building the confidence of HCPs in the safety of the hepatitis B vaccine, and increasing access to vaccination services could improve completion rates.

## Supporting information

**S1 Appendix.**
(DOCX)

**S1 Data.**
(XLS)

## Acknowledgments

Our special thanks go to all healthcare providers who spared their time to participate in the study. Our credit also goes to the research assistants for conducting the data collection process with due diligence. We also remain indebted to the Wakiso district office for the technical support provided to the investigators during the data collection exercise.

## Author Contributions

**Conceptualization:** Tonny Ssekamatte, Zirimala Paul Alex Kimoga, Samuel Etajak, John Bosco Isunju, Solomon Tsebeni Wafula.

**Formal analysis:** Tonny Ssekamatte, Trasias Mukama, Simon P. S. Kibira, Rawlance Ndejjo, Justine Nnakate Bukenya, Zirimala Paul Alex Kimoga, Samuel Etajak, Rebecca Nuwematsiko, Esther Buregyeya, John Bosco Isunju, Richard Kibirango Mugambe, Solomon Tsebeni Wafula, Joan Nankya Mutyoba.

**Methodology:** Tonny Ssekamatte, Trasias Mukama, Zirimala Paul Alex Kimoga, Solomon Tsebeni Wafula.

**Project administration:** Tonny Ssekamatte.

**Supervision:** Tonny Ssekamatte, Zirimala Paul Alex Kimoga.

**Writing – original draft:** Tonny Ssekamatte, Trasias Mukama, Simon P. S. Kibira, Rawlance Ndejjo, Justine Nnakate Bukenya, Zirimala Paul Alex Kimoga, Samuel Etajak, Rebecca Nuwematsiko, Esther Buregyeya, John C. Ssempebwa, John Bosco Isunju, Richard Kibirango Mugambe, Aisha Nalugya, Solomon Tsebeni Wafula, Joan Nankya Mutyoba.

**Writing – review & editing:** Tonny Ssekamatte, Trasias Mukama, Simon P. S. Kibira, Rawlance Ndejjo, Justine Nnakate Bukenya, Zirimala Paul Alex Kimoga, Samuel Etajak, Rebecca Nuwematsiko, Esther Buregyeya, John C. Ssempebwa, John Bosco Isunju, Richard Kibirango Mugambe, Aisha Nalugya, Solomon Tsebeni Wafula, Joan Nankya Mutyoba.

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
