## [Decision Letter · Decision Letter 0]

17 Jun 2020

Hepatitis B screening and vaccination status of healthcare providers in Wakiso district, Uganda

PONE-D-20-13543

Dear Dr. Ssekamatte,

We’re pleased to inform you that your manuscript has been judged scientifically suitable for publication and will be formally accepted for publication once it meets all outstanding technical requirements.

Kind regards,

Isabelle Chemin, PhD

Academic Editor

PLOS ONE

1) Please include a copy of the survey questions or questionnaire used in the study, in both the original language and English, as Supporting Information, or include a citation if it has been published previously. Please state whether you validated the questionnaire prior to testing on study participants. Please provide details regarding the validation group within the methods section.

Reviewers' comments:

Reviewer's Responses to Questions

**Comments to the Author**

1. Is the manuscript technically sound, and do the data support the conclusions?

Reviewer #1: Yes

2. Has the statistical analysis been performed appropriately and rigorously? 

Reviewer #1: Yes

3. Have the authors made all data underlying the findings in their manuscript fully available?

Reviewer #1: Yes

4. Is the manuscript presented in an intelligible fashion and written in standard English?

Reviewer #1: Yes

5. Review Comments to the Author

Reviewer #1: This cross-sectional study was conducted in health care facilities in Wakiso, Uganda to address hepatitis B screening and vaccination among health care providers (HCPs) and associated factors. All results reported were based on self-administered questionnaires.

The methodology was clearly described and the statistical analyzes well conducted.

The authors showed that among the 306 HCPs, 75.2% reported had been screened for hepatitis B and 57.8 % reported to be fully vaccinated. Believing in the efficacy and safety of the vaccine and working in a private healthcare facility were associated with being fully vaccinated. While working in a facility with infection control guidelines, perceiving themselves to be at risk of hepatitis B infection and surprisingly perceiving their job as a risk for infection were less likely to report full vaccination status.

It would be interesting to discuss why “considering oneself to be at risk” is less likely associated with being fully vaccinated in Uganda. This was previously reported in recent publication china.

Limitations of the study:

- Have the health care workers' vaccination cards been randomly checked? or are all reported results based solely on the good faith of health care providers?

-The lack of blood tests (anti-HBs antibodies).

This would have made it possible to know among those who were vaccinated, how many had an antibody level higher than 10 IU.

6. PLOS authors have the option to publish the peer review history of their article (what does this mean?). If published, this will include your full peer review and any attached files.

Reviewer #1: No

---

## [Editor Report · Acceptance letter]

26 Jun 2020

PONE-D-20-13543 

Hepatitis B screening and vaccination status of healthcare providers in Wakiso district, Uganda 

Dear Dr. Ssekamatte:

I'm pleased to inform you that your manuscript has been deemed suitable for publication in PLOS ONE. Congratulations! Your manuscript is now with our production department. 

Kind regards, 

on behalf of

Mrs Isabelle Chemin 

Academic Editor

PLOS ONE